# Could Chronic Idiopatic Intestinal Pseudo-Obstruction Be Related to Viral Infections?

**DOI:** 10.3390/jcm10020268

**Published:** 2021-01-13

**Authors:** Emanuele Sinagra, Gaia Pellegatta, Marcello Maida, Francesca Rossi, Giuseppe Conoscenti, Socrate Pallio, Rita Alloro, Dario Raimondo, Andrea Anderloni

**Affiliations:** 1Gastroenterology and Endoscopy Unit, Fondazione Istituto San Raffaele-Giuseppe Giglio, Contrada Pietra Pollastra Pisciotto, 90015 Cefalù, Italy; fraross76@hotmail.com (F.R.); dottgconoscenti@gmail.com (G.C.); dario.raimondo@hsrgiglio.it (D.R.); 2Euro-Mediterranean Institute of Science and Technology (IEMEST), 90139 Palermo, Italy; 3Digestive Endoscopy Unit, Division of Gastroenterology, Humanitas Clinical and Research Center (IRCCS), 20089 Rozzano, Italy; gaia.pellegatta@gmail.com (G.P.); andrea_anderloni@hotmail.com (A.A.); 4Gastroenterology and Endoscopy Unit, S. Elia-Raimondi Hospital, 93100 Caltanissetta, Italy; marcello.maida@hotmail.it; 5Endoscopy Unit, University Hospital Policlinic G. Martino, 98125 Messina, Italy; socratep@tin.it; 6Division of General and Oncologic Surgery, Department of Surgical, Oncological and Oral Sciences (DICHIRONS), University of Palermo, 90133 Palermo, Italy; ritalloro@hotmail.it

**Keywords:** virus, chronic idiopathic intestinal pseudo-obstruction, gastrointestinal motility disorders

## Abstract

Chronic idiopathic intestinal pseudo-obstruction (CIIPO) is a disease characterized by symptoms and signs of small bowel obstruction in the absence of displayable mechanical obstruction. Due to the known neuropathic capacity of several viruses, and their localization in the intestine, it has been hypothesized that such viruses could be involved in the pathogenesis of CIIPO. The most frequently involved viruses are John Cunningham virus, Herpesviridae, Flaviviruses, Epstein–Barr virus and Citomegalovirus. Therefore, the present narrative review aims to sum up some new perspectives in the etiology and pathophysiology of CIIPO.

## 1. Introduction

Chronic idiopathic intestinal pseudo-obstruction (CIIPO) is a disease characterized by symptoms and signs of small bowel obstruction in the absence of displayable mechanical obstruction [1,2,3].

In a Japanese survey, the prevalence of CIIPO was estimated to be 0.80 to 1.00 per 100,000, with an incidence of 0.21 to 0.24 per 100,000 [4].

It represents one of the most frequent reasons of chronic intestinal failure both in children (15%) and adults (20%) [3,5,6,7,8]. In particular, patients affected by CIIPO are unable to continue with enteric nutrition and consequently to maintain their body weight.

CIIPO is an idiopathic condition in almost fifty percent of cases [4]. However, affected patients need to be investigated by further diagnostic procedures (imaging, endoscopic, laboratory testing, etc.) just to exclude secondary causes of CIIPO. Among these, all the diseases affecting the control mechanisms of intestinal working at different levels (muscular, and intrinsic and extrinsic neural) could be responsible for secondary CIIPO [2]. In fact, the extrinsic autonomic nervous system can be affected both centrally (for example, Parkinson’s disease, vascular stroke, encephalitis, neoplasms and diseases affecting the encephalic autonomous centers) and peripherally (for example, diabetic neuropathy, or further neuropathies involving the enteric nervous system such as Hirschsprung Disease and Chagas Disease) [2,9]. This also includes paraneoplastic syndromes [10,11,12], immune-mediated [13,14] and collagen diseases, and viral infections. Furthermore, radiation enteritis, collagenosis, jejunal diverticulosis, and Ehlers-Danlos syndrome could cause neuronal and myogenic dysfunction [2].

CIIPO is usually sporadic, even if familial forms have also been described [2,3,15,16].

The most common clinical features of CIIPO are represented by bloating, abdominal pain and distension, which may be acute, recurrent, or chronic [3,5].

The diagnosis of CIIPO is usually based on the finding of longstanding symptoms of mechanical obstruction in the absence of an anatomic cause on radiologic examination and endoscopy, and evidence of impaired motility [2]. We can distinguish CIIPO from other causes of mechanical obstruction or further acute functional causes of obstruction based on the time course, location of dilation, symptom progression, and findings on imaging [17,18].

Manometry is expected to represent the gold standard for diagnosis [19], even if wireless motility capsule or the SmartPill methodology [20], through the assessment of both the gastric and colonic function, represents an important adjunct.

Full-thickness small bowel biopsies, acquired in surgical specimens, could help in identifying the site of damage and in guiding therapy [2,19].

In the diagnostic work-up of suspected CIIPO, laboratory tests can help to find secondary causes: complete blood count, electrolytes, liver tests, vitamin B12, folate, celiac serologies, thyrotropin, serologic test for herpes simplex virus (HSV), cytomegalovirus (CMV), Epstein–Barr virus (EBV), markers of inflammation (e.g., erythrocyte sedimentation rate-ESR, C-reactive protein-CRP).

In particular, searching for the aforementioned viral infections is mandatory because several systemic viral infections could involve the enteric nervous system with the consequence of an alterated peristaltic activity [1]. The most involved viruses are John Cunningham virus (JCV), HSV, EBV, CMV, varicella zoster virus (VZV) and Flaviviruses.

Therefore, the present narrative review aims to sum up some new perspectives in the etiology and pathophysiology of CIIPO.

## 2. JC Virus

JCV belongs to human polyomaviruses and to the Papovaviridae family and is a non-enveloped, small (45 nm) virus with a circular double-stranded DNA genome [21,22].

JCV was firstly identified in 1952 in a patient (with initials “J.C.”) affected by Hodgkin’s lymphoma and progressive multifocal leukoencephalopathy (PML) [21,23]. It can be transmitted through fecal–oral and respiratory routes [24,25], blood transfusions, organ transplantation, trans-placentally and seminal fluid [21,26,27]. After acute infection, JCV enters a latent phase in several human tissues. For example, JCV can be detected in tonsilar tissues of 39% of healthy individuals [21,22,28].

JCV is most commonly known as being associated with PML in immunosuppressed patients [29] and in particular in 5% of patients affected by Acquired Immune Deficiency Syndrome (AIDS) [21,30].

In fact, in literature, two possible mechanisms of cell infection by JCV are described [29]. The pathogenic mechanism described in cells capable to replicate viral DNA is a lytic infection with viral amplification such as PML [29]. In non-permissive cells that do not permit viral replication, the mechanism is an abortive infection or cell malignant transformation [30,31], as reported for colon and lung cancer [31,32,33,34,35,36]. However, a potential role of a chronic JCV infection in CIIPO pathogenesis has been hypothesized in observational studies [37,38].

Selgrad et al., performed a case-control study comparing specimens from 10 patients with CIIPO with 61 control specimens. In this study, JCV proteins (TAg and VP1) were identified in glial cells of the myenteric plexus in seven out of 10 adult patients with idiopathic CIIPO and in none of the control specimens [37].

TAg and VP1 were identified in the myenteric plexuses, indicating a lytic infection, potentially destroying the host cell; therefore, virus detection in enteroglial cells leads to a possible pathological role for JCV in enteric neuropathy.

Our group also reported two cases of CIIPO associated with JCV infection [38] and virus DNA was detected by polymerase chain reaction (PCR) amplifications in duodenal and jejunal samples [38].

## 3. Herpes Virus

Herpes viruses belong to a large family of viruses which present similar structural characteristics such as a linear double stranded DNA genome, envelope, capsid and tegument.

Herpes viruses are divided into three subgroups: α (alfa) which includes herpes simplex virus 1 (HSV-1), herpes simplex virus 2 (HSV-2) and varicella zoster virus (VZV); β (beta) cytomegalovirus (CMV), human herpes type 6 (HHV-6) and human herpesvirus type 7 (HHV-7) and γ (gamma) Epstein-Barr virus (EBV) and human herpes type 8 (HHV-8) [39,40].

All herpes viruses are neurotropic and are able to cause motility disorders. However, to date, cases of colonic and ileal pseudo-obstruction have been reported in the literature only associated with VZV, CMV and EBV infection.

## 4. VZV

Primary infection with VZV causes chickenpox which occurs with fever and diffuse pruritic rash usually affecting children.

After the acute infection, the virus enters a latent phase in nerves, specifically in the dorsal ganglion cells of sensory nerves, the cranial nerves and in the enteric nervous system (ENS) ganglia [41,42]. This is called the life-long latency period.

VZV can reach the ENS through transport by T lymphocytes during the acute infectious phase or with a retrograde transport from the epidermal projections of infected dorsal-root ganglion neurons to ENS [41,42].

Endogenous viral re-activation with VZV replication and ganglion inflammation occurs in about 30% of subjects also many years after the primary infection [41]. Factors predisposing to virus reactivation are advanced age, malnutrition, stress, neoplasms, menstruation, chemotherapy.

During this phase, the virus moves antegrade from the ganglia along the axons of the neurons to the skin causing the so-called Herpes Zoster. In addition, in about 5% of subjects the virus reactivation can lead to peripheral somatic and visceral motor neuropathies. Moreover, based on the location of virus infection (i.e., diaphragm, upper and lower limb muscles, trunk, bladder and gut), patients can present variable clinical manifestations [43,44,45].

Gastrointestinal symptoms associated with VZV reactivation are very rare; however, some case reports have demonstrated the coexistence of VZV and colonic pseudo-obstruction [46,47,48,49,50,51,52,53,54,55,56,57,58,59,60,61]. Colonic pseudo-obstruction caused by VZV was first reported in a 1950 case report.

On the contrary, only a few cases are reported of paralytic ileus (PI) following VZV infection [49,62,63,64,65,66].

Typical symptoms of ileal and colonic pseudo-obstruction (abdominal pain, distension, constipation) usually occur a few days or weeks before the rash, making a correct diagnosis very difficult. The most accurate test for the diagnosis of Herpes Zoster is PCR for VZV DNA in colonic mucosal biopsy samples [46]. Only two studies reported the presence of VZV in small bowel muscle specimens in patients affected by paralytic ileus [46,67] and only one study reported the presence of VZV in the myenteric plexus [65].

Different hypotheses on pathogenesis of ileal and colonic pseudo-obstruction due to VZV have been formulated: (1) direct viral injury to colonic ENS and muscularis propria; (2) direct infection of celiac plexus ganglion and colonic autonomic nervous system (ANS) [67]; (3) viral involvement of the parasympathetic nerves (e.g., thoracolumbar or sacral lateral columns) causing intestinal hypomotility [68]; (4) hemorrhagic infarction of abdominal sympathetic celiac ganglia [67]; (5) peritoneal inflammation due to vescicular eruptions [56]; (6) injury of afferent C-fibers leading to intestinal pseudo-obstruction [69].

Figure 1 shows the pathogenesis of VZV infection.

## 5. CMV

Cytomegalovirus (CMV) is a member of herpes virus family and is widespread in the world population.

CMV infection of the digestive tract is usually seen in immunodeficient patients and are characterized by ulcerations, erosions, and mucosal hemorrhage. A case of CIIPO was reported in an immunocompromised patient who had previously undergone a cardiac transplant [70]. CMV inclusions have also been identified in the myenteric plexus of the intestine and colon of patients who have previously undergone renal transplantation [71].

However, since 1964, several studies have reported GI disease due to CMV in patients without detectable immunodeficiency [72]. To date, some case reports and a miniseries of cases in which CIIPO has been attributed to congenital and infant CMV infection have been published.

Dèchelotte et al. reported three cases of paralytic ileus in fetuses caused by CMV infection. In these cases, CMV had been identified in the ganglion cells or in the myenteric and submucosal plexuses of the small bowel and colon.

In one patient, paralytic ileus was transient and associated with a mild CMV infection not even identified at autopsy. In the two other cases reported, paralytic ileus persisted and was associated with severe CMV infection with inflammatory reaction and cell necrosis [73].

Foucaud et al. reported a case of a 2-month-old boy affected by abdominal distension and inappetence leading to total parenteral feeding. Viruria and specific IgM and IgG antibodies confirmed CMV infection. The patient also underwent rectal biopsy with detection of hypoganglionosis, nerve thickening and CMV intranuclear inclusions [74].

Moreover, a case of a 2-month-old girl affected by CMV-related CIIPO with abdominal distension and vomiting has been reported. At the small bowel biopsy, CMV was identified. Moreover, the patient’s trend of IgM and IgG and the absence of maternal IgG were indicative of infection within the first few weeks of life [75].

Debinsky et al. evaluated the evidence of viral DNA in samples from 13 adult patients affected by CIIPO and 12 controls. Among CIIPO patients, three were positive; in particular, one patient, who presented visceral neuropathy and myopathy, had small bowel biopsies positive for CMV DNA. In control tissue, any virus was detected [76].

## 6. EBV

Epstein-Barr virus (EBV) infection, the so-called infectious mononucleosis, frequently involves the nervous system [77] with cases of acute pandysautonomia and CIIPO described in literature [78].

Two pathogenetic mechanisms have been hypothesized: (1) viral injury of autonomic ganglia or postganglionic neurons [78,79]; (2) abnormal T-cell- and B-cell-mediated immune response with damage to neuronal tissue cross-reacting with a viral antigen [80].

Vassallo et al. reported the first case of selective cholinergic dysautonomia that occurred following acute infectious mononucleosis. In particular, they reported the case of a young woman affected by nausea, vomiting, severe abdominal distention and constipation, insurgents about 3 years after an episode of EBV acute infection. Also, pupils, sweat glands, lacrimal and salivary glands, and urinary bladder were affected. Injury of sympathetic and parasympathetic postganglionic cholinergic nerves with the preservation of sympathetic adrenergic functions was identified at autonomic function tests. In addition, the ganglion cells of the myenteric plexus were normal at morphological and immunohistochemical evaluation and therefore the selective cholinergic dysnfuction was identified as the most likely cause of CIIPO [81].

Besnard et al. published the case of a little boy affected by pharyngitis who subsequently developed intestinal obstruction, pandysautonomia and encephalomyelitis. After 3 months from the first onset of symptoms, the patient has undergone resection appendix and sigmoidoscopy and the tissue samples showed hypoganglionosis and a mononuclear inflammatory infiltrate in the myenteric neural plexus. EBV-RNA was detected in the blood and cerebrospinal fluid, and EBV-RNA was identified in myenteric appendix cells, in a mesenteric lymph nodeand in gastric biopsies. The serology for EBV showed previous infection but anti-early antigen antibodies were present [82].

Two of the three CIIPO patients who tested positive in the Debinsky et al., series presented EBV DNA. Specifically, in a patient with visceral neuropathy, EBV DNA was isolated in the small intestine, while the other patient with visceral myopathy had EBV DNA in both the colon and the small bowel [76].

Finally, De Giorgio et al., performed a PCR-based study of 13 patients with CIIPO diagnosticated clinically and manometrically. The study revealed an incidence of 31% of positive patients, with two affected by HSV-1 and two by EBV infections [1].

Figure 2 shows the pathogenesis of EBV infection.

## 7. Flavivirus

Flavivirus belongs to a group of RNA virus that include the West Nile virus (WNV), Zika virus (ZIKV), Powassan virus (POWV), dengue virus, tick-borne encephalitis virus, yellow fever virus, and several other viruses. These viruses have some similar characteristics: size (40–65 nm), an icosahedralnucleocapsid, positive-sense, single-stranded RNA, and similar appearance in the electron microscope.

In particular, WNV is a Flavivirus that is transmitted via mosquitoes and causes an acute fever [83]. It is a neurotropic virus which leads to injury in the cerebral cortex, brain stem and spinal cord causing meningitis, encephalitis and even death in infected subjects [84].

ZIKV is another virus belonging to this genus and mostly causes congenital malformations in fetuses during the period of pregnancy [85].

To date, there has been a lack of data published in literature regarding the infection of Flavivirus on ENS neurons and consequent gastrointestinal symptoms.

Some studies have been conducted on mice and have shown that mice infected can develop gastrointestinal pathology [86,87]. In confirmation of this, viral antigens, RNA, lesions of the myenteric plexus and necrosis of enterocytes have been identified in the intestinal tissue isolated in infected rodents [86,87].

A recently published study showed that mice infected with WNV or Kunjin virus, during the acute phase, present elevated levels of viral RNA associated with intestinal dilation, damage of ENS neurons and CD8^+^T cell intestinal infiltration. The gradual clearance of viral RNA corresponds to an improvement of intestinal transit and an inflammatory infiltrate reduction [88].

Moreover, mice with persistence of WNV or KUNVRNA in the intestine suffer from chronic GI transit delay that can last for weeks and months after the acute phase and are prone to GI symptom exacerbations [89].

To date, the few data in humans are case reports of patients affected by Flavivirus acute infection and GI symptoms (e.g., vomiting, diarrhea and abdominal pain) [89,90]. The virus was also identified in the intestinal tissue of patients with WNV acute infection [91].

## 8. Discussion

The management of individuals affected by CIIPO is often challenging [92].

In its management, the most relevant diagnostic step is to exclude secondary causes susceptible to treatment.

Due to the known neuropathic ability of the aforementioned virus, and their frequent identification in the intestine, it has been hypothesized that such viruses could be identifiable in tissues of patients with CIIPO, and be involved in the etiology of this condition. The studies reported in this review, although represented by case reports or case series, suggest that CIIPO may be caused by a viral infection through the alteration of the integrity of the ENS.

In Table 1, the evidence regarding the interlacing between CIIPO and viral infections is summarized.

Despite the fact that serology could be helpful for diagnosis as it shows a predictive negative value, only full-thickness biopsy of the intestine, as shown for VZV [46,47], CMV [74,75,76], EBV [76,81], JCV [37,38], together with the help of PCR-based studies may orient the diagnosis of a direct effect of a viral infection on intestine.

However, we should also consider that sequencing viral nucleic acids, whether from cultures or directly from clinical specimens, is complicated by the presence of contaminating host DNA [93]. Therefore, sometimes the detection of viral DNA or RNA (a fragment of genome in most cases) is not correlated with viral expression.

Currently, genome sequencing of viruses can be achieved by ultra-deep sequencing or through the enrichment for viral nucleic acids before sequencing, either directly or by concentrating virus particles, but these approaches have their own costs and complexities [93].

Finally, the pivotal action of the ENS in certain diseases of the gastrointestinal tract, and the identification of receptors on the enteric neuron for hormones and transmitters, suggests the rational basis for current and future targeted therapies potentially helping patients suffering from a broad range of gastrointestinal disorders [94].

Some reports showed that immunomodulatory/immunosuppressive therapies were effective in CIIPO with an inflammatory neuropathy, if antineuronal antibodies are detected (ANNA-1 or anti-Hu), or in autoimmune-induced CIIPO [19,95,96].

Therefore, therapies promoting gastric and colonic motility will be the future optimal therapies to promote gut and small bowel transit. For example, neurotrophins could help to treat enteric neuronal degeneration, since these proteins stimulate the growth of nerve cells [19]. In fact, as a potential therapy in the event of proven viral involvement, neurotrophins have been shown to improve colonic transit and resolve constipation; they are also involved in maintenance programs in adult enteric neurons through a role in antioxidant defense [97]. In this setting, as previously suggested [19], BCL-2 could potentially be employed as a marker for neuronal survival in patients with a presumed degenerative enteric neuropathy [19].

In spite of the fact that good evidence is scanty, it is intriguing to consider that CIIPO could be the consequence of an interaction with the several luminal microorganisms that populate the gut wall [98].

## Figures and Tables

**Figure 1 jcm-10-00268-f001:**
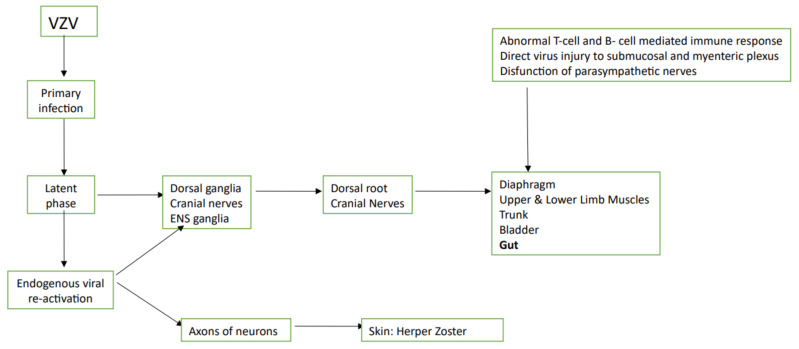
Pathogenesis if varicella-zoster infection is present. ENS: enteric nervous system; VZV: varicella-zoster virus.

**Figure 2 jcm-10-00268-f002:**
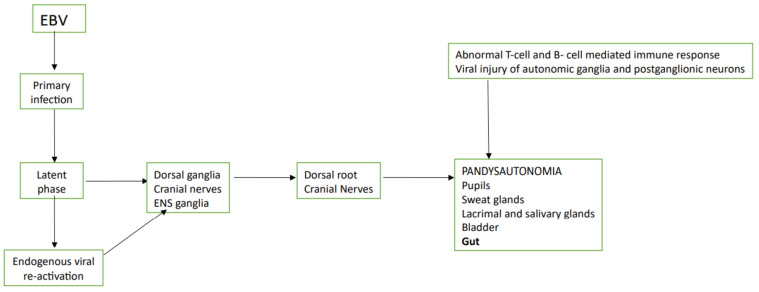
Pathogenesis if Epstein-Barr virus is present. ENS: enteric nervous system; EBV: Epstein-Barr virus.

**Table 1 jcm-10-00268-t001:** Summary of the evidence regarding the interlacing between chronic idiopatic intestinal pseudo-obstruction and viral infections.

Reference	Author	Year of Publication and Study Protocol	Virus Involved and Technique of Its Detection	Suspected Mechanism of Action
[37]	Selgrad, et al.	2009, case-control study	John Cunningham Virus. DNA was extracted from the myenteric plexuses of colonic and ileal specimens, and JCV T antigen (TAg) DNA and the viral regulatory region were detected by PCR and sequencing	The John Cunningham Virus localization in enteroglial cells suggests a possible pathological role for this virus in enteric neuropathy
[38]	Sinagra, et al.	2020, case report on 2 patients	John Cunningham Virus. PCR amplifications were performed using gene-specific primers for T antigen; PCR for the only JCV was positive in duodenal and jejunal samples in both the patients	The John Cunningham localization in small bowel suggested a possible pathological role for this virus in enteric neuropathy not otherwise classified
[62]	Johnson, et al.	1977, case report	Varicella Zoster Virus. Detection through PCR in ileocolonic samples	Direct viral injury to colonic enteric nervous system and muscularispropria
[46]	Carrascosa, et al.	2014, case report	Varicella Zoster Virus. Detection through PCR in colonic samples	Direct viral injury to colonic enteric nervous system and muscularispropria
[63]	Cane, et al.	1959, case report	Varicella Zoster Virus. First detection in ileal samples	Direct viral injury to colonic enteric nervous system and muscularispropria
[64]	Hiramatsu, et al.	2013, case report	Varicella Zoster Virus. Detection through PCR in ileal samples	Direct viral injury to colonic enteric nervous system and muscularispropria
[65]	Chang, et al.	1978, case report	Varicella Zoster Virus. Detection through PCR in ileal samples	Direct viral injury to colonic enteric nervous system and muscularispropria
[66]	Anaya-Prado, et al.	2018, case report	Varicella Zoster Virus. Detection through PCR in ileocolonic samples	Direct viral injury to colonic enteric nervous system and muscularispropria
[67]	Pui, et al.	2001, case report	Varicella Zoster Virus. Detection through PCR in ileocolonic samples	Direct infection of celiac plexus ganglion and colonic autonomic nervous system (ANS) and hemorrhagic infarction of abdominal sympathetic celiac ganglia
[68]	Tribble, et al.	1993, case report	Varicella Zoster Virus. Detection through PCR in ileocolonic samples	Viral involvement of the thoracolumbar or sacral lateral columns causing parasympathetic nerves disfunction and intestinal hypomotility
[56]	Nomdedéu, et al.	1995, case report	Varicella Zoster Virus. Detection through PCR in ileocolonic samples	Peritoneal inflammation due to vescicular eruptions
[70]	Hosoe, et al.	2010, case report	Varicella Zoster Virus. Detection through PCR in colonic samples	Injury of afferent C-fibers leading to intestinal pseudo-obstruction
[74]	Dèchelotte, et al.	1992, case report on 3 patients	Cytomegalovirus. Virus had been identified through PCR in the ganglion cells or in the myenteric and submucosal plexuses of the small bowel and colon	Antenatal paralytic ileus caused by cytomegalovirus infection
[75]	Foucaud, et al.	1985, case report	Cytomegalovirus. Viruria and specific IgM and IgG antibodies confirmed cytomegalovirus infection. The patient underwent also rectal biopsy with detection of hypoganglionosis, nerve thickening and cytomegalovirusintranuclear inclusions	Paralytic ileus caused by cytomegalovirus infection
[77]	Debinsky, et al.	1992, case control study	Cytomegalovirus. One patient, who presented visceral neuropathy and myopathy, had small intestine samples positive for cytomegalovirus DNA. No control tissue was positive for any virus	Visceral neuropathy and myopathy caused by cytomegalovirus
[76]	Ategbo, et al.	1996, case report	Cytomegalovirus. At the small bowel biopsy, cytomegalovirus was identified. Moreover, the patient’s trend of IgM and IgG and the absence of maternal IgG were indicative of infection within the first few weeks of life	Visceral neuropathy caused by cytomegalovirus
[82]	Vassallo, et al.	1991, case report	Epstein–Barr virus. The ganglion cells of the myenteric plexus were normal at morphological and immunohistochemical evaluation and therefore the selective cholinergic dysautonomia was identified as the most likely pathophysiologic process responsible for the symptoms	Selective cholinergic dysautonomia that occurred following acute infectious mononucleosis
[83]	Besnard, et al.	2000, case report	Epstein–Barr virus. The patient has undergone resection appendix and sigmoidoscopy and the tissue samples showed hypoganglionosis and a mononuclear inflammatory infiltrate in the myenteric neural plexus. EBV-PCR was positive in the blood and cerebrospinal fluid, and EBV-RNA was identified in myenteric appendix cells, in a mesenteric lymph node, and in gastric biopsies	Viral direct invasion of autonomic ganglia or postganglionic neurons
[1]	De Giorgio, et al.	2010, case control study	Epstein–Barr virus. PCR-based study of 13 clinically and manometrically characterized patients with chronic idiopatic intestinal pseudo-obstruction (2 patients positive for Epstein–Barr virus infection)	Viral direct invasion of autonomic ganglia or postganglionic neurons
[92]	Armah, et al.	2007, post-mortem study	West Nile Virus, Flavivirus acute infection affected by GI symptoms and located in the intestine	Viral antigens, RNA, lesions of the myenteric plexus and necrosis of enterocytes have been identified in the intestinal tissue isolated also in infected rodents

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
