# Peer review of "Could Chronic Idiopatic Intestinal Pseudo-Obstruction Be Related to Viral Infections?"

_jcm, 2021, doi:10.3390/jcm10020268_

Round 1
Reviewer 1 Report
Dear Authors,
thank you for your submission.
It is an interesting paper exploring the role of viral infection in the development of CIIPO. The description of the viruses involved is well structured. I would like, if possible, to know in more details what do the authors think about potential therapy options for CIIPO in the event of proven viral involvement.
Reviewer 2 Report
To my opinion, the role of viruses on chronic idiopathic intestinal pseudo-obstruction is not so clear and is not frequent. The pathological role of the viruses should be more explained (even if these aue only hypothetical). The authors recommend serology for diagnosis: this method can not prove the direct effect on intestine. And as almost 100% of adult population have already been infected by JCv, how can you explain its role? For CMV, the authors give some data with direct methods for CMV infection detection.
Table(s) and figure(s) should be provided.
Reviewer 3 Report
In this narrative review the authors provide some new perspectives with regard to the etiology and pathophysiology of chronic idiopathic intestinal pseudo-obstruction, with a focus on possible influences of viral infections.
The manuscript is generally well written and gives an overview of possibly involved viruses and arguments supporting these hypotheses.
The manuscript would benefit from a table summarising all these data, thereby increasing the global overview of the data and supporting arguments.
On line 227, KUNV appears in the manuscript. This should first be mentioned as "Kunjin virus (KUNV)".
Finally, I recommend proofreading by a native English speaker.
Round 2
Reviewer 1 Report
Dear Authors
the paper is interesting
Good work
Author Response
Dear Reviewer,
We would like to express our appreciation to You for Your insightful comments, which have helped us significantly to improve our manuscript.
Sincerely Yours
Emanuele Sinagra
Reviewer 2 Report
I would like to thank the authors for improving their manuscript with figures and tables and adding information. Just one last remark: the detection of viral DNA or RNA (a fragment of genome in most cases) is not correlated with viral expression; the affirmation of cause/effect relation should be modulated.
Author Response
Dear Reviewer,
We would like to express our appreciation to You for Your insightful comments, which have helped us significantly to improve our manuscript. According to the suggestion, we have thoroughly revised our manuscript and its final version is enclosed. Point-by-point responses to the comments are listed below
Reviewer 's remark #2
I would like to thank the authors for improving their manuscript with figures and tables and adding information. Just one last remark: the detection of viral DNA or RNA (a fragment of genome in most cases) is not correlated with viral expression; the affirmation of cause/effect relation should be modulated.
Response:
We modified the conclusion section as following, considering that sometimes the detection of viral DNA or RNA (a fragment of genome in most cases) is not correlated with viral expression, therefore the affirmation of cause /effect relation should be modulated according also to the technique used for detection:
"However, we should also consider that sequencing viral nucleic acids, whether from cultures or directly from clinical specimens, is complicated by the presence of contaminating host DNA. Therefore, sometimes the detection of viral DNA or RNA (a fragment of genome in most cases) is not correlated with viral expression.
Currently, genome sequencing of viruses can be achieved by ultra-deep sequencing or through the enrichment for viral nucleic acids before sequencing, either directly or by concentrating virus particles, but these approaches have their own costs and complexities."
Many thanks again,
Sincerely
Authors
This manuscript is a resubmission of an earlier submission. The following is a list of the peer review reports and author responses from that submission.